# Nitric Oxide Contributes to the Pathogenesis of *Nocardia farcinica* Infection in BALB/c Mice and Alveolar MH-S Macrophages

**DOI:** 10.3390/microorganisms13102336

**Published:** 2025-10-10

**Authors:** Jiang Yao, Lichao Han, Jirao Shen, Bingqian Du, Ziyu Song, Min Yuan, Shuai Xu, Xiaotong Qiu, Xueping Liu, Fang Li, Yihe Liang, Wanchun Guan, Zhenjun Li

**Affiliations:** 1Wenzhou Key Laboratory of Sanitary Microbiology, Key Laboratory of Laboratory, Medicine, Ministry of Education, School of Laboratory Medicine and Life Sciences, Wenzhou Medical University, Wenzhou 325035, China; yaojiang0919@gmail.com (J.Y.); songziyu0923@gmail.com (Z.S.); 2National Key Laboratory of Intelligent Tracking and Forecasting for Infectious Diseases, National Institute for Communicable Disease Control and Prevention, Chinese Center for Disease Control and Prevention, Beijing 102200, China; han787974996@163.com (L.H.); shenjirao0828@gmail.com (J.S.); 2112041038@stu.gdpu.edu.cn (B.D.); yuanmin@icdc.cn (M.Y.); xushuai@icdc.cn (S.X.); qiuxiaotong@icdc.cn (X.Q.); liuxueping8712@163.com (X.L.); lifang_2021@163.com (F.L.); liangyihe2016103@163.com (Y.L.)

**Keywords:** *Nocardia farcinica*, nitric oxide, pathogenesis, alveolar macrophages

## Abstract

*Nocardia*, a rare but life-threatening pathogen, can invade multiple tissues and organs, such as lungs, brain, skin and soft tissue. In this study, we determined whether nitric oxide (NO) contributes to the severity of experimental pulmonary nocardiosis. BALB/c mice with or without aminoguanidine (AG) treatment were infected with *N. farcinica* through intranasal or intraperitoneal routes. Over experimental period, weight and mortality were monitored, and lung tissues were collected for NO production, cytokines detection, histopathological analysis, and bacterial load assessment. Next, alveolar MH-S macrophages were treated with various inhibitors to explore the impacts of NO, MAPK, and NF-κB against *N. farcinica* infection. AG treatment improved weight loss, lowered pulmonary bacterial load, and attenuated inflammatory response in infected mice. Similar effects were observed in alveolar MH-S macrophages. And all AG-treated mice survived infection. Furthermore, we suggest that NO is induced by *N. farcinica* through MAPK JNK and NF-κB signaling. Our study demonstrates the causative role of inducible NO on the severity of *N. farcinica* infection.

## 1. Introduction

*Nocardia* is a Gram-positive pathogen, often resulting in pneumonia, brain abscess, and so forth [1,2]. Recently, with increasing number of the elderly and immunosuppressed individuals, and progression of detection technology, the incidence rate of nocardiosis has grown [3,4]. As inhalation is the main route of exposure to *Nocardia*, pulmonary nocardiosis is the most common type of infection [5,6]. If pulmonary nocardiosis is not controlled timely, it is more likely to disseminate to other parts of body and cause worse prognosis.

Currently, 130 *Nocardia* species have been effectively characterized and correctly named (https://lpsn.dsmz.de/genus/nocardia, accessed on 30 September 2025), 54 of which are related to human infection [7,8,9,10], and *Nocardia farcinica*, *Nocardia cyriacigeorgica*, *Nocardia brasiliensis*, and *Nocardia otitidiscaviarum* are the major etiological factor of nocardiosis in human [11]. In China, *N*. *farcinica* is the most frequently isolated *Nocardia* species and primarily causes lung and disseminated nocardiosis [11,12,13]. There is an urgent need to explore the pathogenesis of *N*. *farcinica* due to its wide distribution, invasiveness, dissemination capacity [11].

Some reports have elucidated possible pathogenic mechanisms of *Nocardia*. Mammalian cell entry (Mce) proteins 1E and Nfa34810 protein of *N*. *farcinica* may facilitate the strain’s invasion of Hela cells [14,15]. Additionally, *N*. *farcinica* was used to infect BV2 cells, with total RNA collected at 2 h post-infection (hpi) to analyze the expression of inducible nitric oxide synthase (iNOS). Results showed that the mRNA level of iNOS was upregulated after infection [4]. However, the association between the upregulation of iNOS and neurological symptoms caused by *N*. *farcinica* was not verified. Carmona et al. investigate the effect of inducible nitric oxide (NO) produced by iNOS on experimental *N. brasiliensis* infections in mice. The data suggested significant upregulation of NO in plasma after infection. Suppression of NO by aminoguanidine (AG), a selective inhibitor of iNOS, protected mice from actinomycetoma development [16]. However, the effect of inducible NO on experimental *N*. *farcinica* infection remains unclear.

Innate immunity is the body’s first line of defense against invading pathogens. Alveolar macrophages are the critical components of respiratory innate immune system. To encounter invading pathogens, macrophages can activate mitogen activated protein kinase (MAPK)/nuclear factor-kappa B (NF-κB) signaling to induce iNOS protein and its product inducible NO expression [17]. Previous pathogenic studies on *N*. *farcinica* have not focused on the effect of MAPK/NF-κB signaling on the expression of NO.

In our study, we constructed experimental cell and mouse models with low NO expression to evaluate the effects of inducible NO on *N*. *farcinica* pulmonary infection. Then, we explored how *N*. *farcinica* regulates inducible NO expression. Our findings suggest that NO is induced by *N. farcinica* through MAPK JNK/NF-κB signaling in alveolar macrophages. Inducible NO is shown to contribute to the pathogenesis of *N. farcinica* infection in alveolar macrophages and mice, offering potential prevention and control strategies for nocardiosis.

## 2. Materials and Methods

### 2.1. Infection of Mice

Female BALB/c mice (SPF Biotechnology Co., Ltd., Beijing, China), 6–8 weeks old, were given 5 days for acclimation after arrival. The mice were kept under a 12 h light–dark cycle and were provided rodent feed (SPF Biotechnology Co., Ltd., Beijing, China) and filtered tap water ad libitum in a sterile environment. All animal experiments were approved by the ethics review committee of the Chinese Center for Disease Control and Prevention, and the ethical approval number was 2023-037.

*N. farcinica* IFM 10152 was purchased from the German Resource Centre for Biological Materials. Before infection, *N. farcinica* IFM 10152 was inoculated into Brain HeartInfusion (BHI) broth (Oxoid Ltd., Hants, UK) and cultivated overnight at 37 °C with continuous shaking at 180 rpm. The subsequent overnight culture was diluted with BHI at 1:100 and incubated at 37 °C with shaking until reaching logarithmic growth phase. Subsequently, the final cultures were subjected to centrifugation, followed by washing, resuspension, and adjustment in phosphate-buffered saline (PBS) [Thermo Fisher Scientific, Waltham, MA, USA] to achieve concentrations of bacterial suspensions (3–6) × 10^7^ colony-forming units (CFU) and 1 × 10^8^ CFU in 50 and 100 µL, respectively.

For inhibition of NO production, mice were injected intraperitoneally with a single dose of 100 μL of AG (100 mg/kg body weight, Sigma-Aldrich, St. Louis, MO, USA) for 7 consecutive days based on a previous report [18] with partial modifications. Equal volume of phosphate-buffered saline (PBS) served as controls. The study by MacFarlane et al. [18] involved administering AG-drinking water to mice for 7 days before infection and continuing until the mice were sacrificed. The 7-day pre-treatment period with AG is intended to allow sufficient time for the drug to achieve effective plasma and tissue concentrations, thereby modulating the host immune response robustly at the time of infection. Meanwhile, we considered that the daily water intake might vary between individual mice and was an uncontrollable factor. Therefore, we switched to the intraperitoneal injection and did not continue administration after infection. For inflammatory studies, PBS-treated and AG-treated mice (6–8 mice per group) were infected intranasally with (3–6) × 10^7^ CFU of *N. farcinica* diluted in PBS in a total volume of 50 µL under anesthesia. For survival experiments, PBS-treated and AG-treated mice (10 mice per group) were given with 1 × 10^8^ CFU of *N. farcinica* diluted in PBS in a total volume of 100 µL by intraperitoneal injection, and mortality was recorded for 14 days.

### 2.2. Record of Body Weight Change

Change in body weight was serially measured for 7 days. Mice were weighed to a thousandth of a gram accuracy using an electronic balance (YH-C2003, Yingheng Intelligent Equipment Co., Ltd., Dongyang, China).

### 2.3. Determination of Bacterial Burden, NO, and Cytokine Concentration in Lung Tissues

After mice were sacrificed by cervical dislocation, whole lung tissues were collected and homogenized in 1 mL of ice-cold PBS for 10 min using a mechanical homogenizer (QIAGEN TissueLyserII, QIAGEN, Duesseldorf, Germany). After homogenization, we checked every sample to ensure that no large pieces of lung tissue were present. We chose to homogenize the entire lung to ensure a representative measurement of the total bacterial load and inflammatory response across the entire organ. While we acknowledge that some studies use individual lobes, our method is consistent across all experimental groups, allowing for valid comparative interpretations. The lung homogenate was serially diluted and plated on BHI agar plates to count bacterial burden. Supernatants from the lung homogenate were used to measure NO (Beyotime Biotechnology Co., Ltd., Shanghai, China), and inflammatory cytokine tumor necrosis factor-alpha (TNF-α, BD OptEIATM, San Diego, CA, USA), interferon-gamma (IFN-γ, Invitrogen, Carlsbad, CA, USA) and interleukin-10 (IL-10, BD OptEIATM, San Diego, CA, USA) concentration under the manufacturer’s instructions.

### 2.4. Lung Histopathology

Lung tissue was fixed in 4% paraformaldehyde (Wuhan Servicebio Technology Co., Ltd., Wuhan, China) and embedded in paraffin. Then, it was cut into 5 mm sections, stained using hematoxylin and eosin and viewed under a biological microscope (Nikon, Tokyo, Japan) according to the manufacturer’s instructions.

### 2.5. Cell Culture

Murine MH-S alveolar macrophage cells (National Infrastructure of Cell Line Resource, Beijing, China) were cultured in Roswell Park Memorial Institute (RPMI) 1640 medium (Gibco, Grand Island, NY, USA) containing 10% fetal bovine serum (FBS; Gibco, Grand Island, NY, USA) at 37 °C in a humidified incubator with 5% CO_2_.

### 2.6. Determination of NO, TNF-α, IFN-γ and Bacterial Replication

Alveolar MH-S macrophages were cultured overnight in a 24-well plate with an initial density of 2 × 10^5^ cells per well. Prior to infection with *N. farcinica*, the cells were treated with or without inhibitors of extracellular regulated protein kinases (ERK) (PD 98059, Sigma-Aldrich, St. Louis, MO USA), c-Jun N-terminal kinase (JNK) (SP 600125, Sigma-Aldrich, St. Louis, MO, USA), p38 (SB 203580, Sigma-Aldrich, St. Louis, MO, USA), or NF-κB (Pyrrolidinedithiocarbamate ammonium, PDTC, APExBIO, Houston, TX, USA), and iNOS (AG) for one hour (h). At 12 hpi, the cell supernatants were collected and used to measure NO, TNF-α, and IFN-γ concentration by Griess reaction and ELISA according to the manufacturer’s instructions. For bacterial replication, the cells were infected with *N. farcinica* in the presence or absence of AG. At the end of scheduled experiments, all cells were lysed with 1 mL H_2_O and plated on BHI agar plates following serial dilution. The bacterial counts were recorded after incubation for 48 h.

### 2.7. Statistical Analysis

All statistical analyses were performed using GraphPad Prism version 9.0.0 software. Survival was analyzed using Log-rank test. Group means and standard deviations were analyzed by Student’s *t*-test, ANOVA, or Two-way ANOVA. The results were significant when the *p*-value < 0.05. Group means, and standard deviations were analyzed by Student’s *t*-test, ANOVA, or Two-way ANOVA.

## 3. Results

### 3.1. NO Results in N. farcinica-Induced Acute Pulmonary Disease In Vivo

Our previous study [19] showed that the bacterial load in lung tissue is higher than that in other organs (e.g., heart, liver, spleen, kidney, eye, brain, rectum) at 7 days after intranasal infection, suggesting almost no dissemination. Thus, to preliminarily evaluate the effect of NO on *N. farcinica* infection, we assessed body weight changes, pulmonary bacterial burden, and release of NO within 7 days after infection. As shown in Figure 1A, weight of infected mice decreased to 1 dpi and then recovered gradually. The initial weight was attained at 7 dpi. While in the control group, it fluctuated slightly throughout the experiment. Pulmonary bacterial burden peaked at 1 dpi (Figure 1B). The concentration of NO in lung tissues enhanced sharply at 1 dpi, then dropped, and approached the control level at 7 dpi (Figure 1C). Accordingly, we assessed disease severity at 1 dpi in subsequent experiment.

Next, mice were treated with AG for 7 consecutive days before infection. AG treatment significantly suppressed, albeit not completely, NO production in infected mice (Figure 2B). Analysis of NO production confirmed the efficacy of AG. We next determined the effect of AG treatment on *N. farcinica*-infected mice. Relative to PBS-treated infected mice, AG-treated infected mice demonstrated significantly less body weight loss (Figure 2D). We then turned to analyze whether inhibiting NO expression by AG treatment would affect bacterial replication during *N. farcinica* infection. Pulmonary bacterial burden has been shown to be lower in AG-treated infected mice than PBS-treated infected mice (Figure 2C). Throughout the entire experiment, no mortality occurred in the uninfected group due to pre-treatment with PBS or AG.

With regard to pulmonary inflammation, pathological examination (hematoxylin and eosin (H&E) staining) on lung sections was performed. Interestingly, lungs from AG-treated infected mice displayed a reduction in number of inflammatory cells relative to PBS-treated infected mice (Figure 3A). Little evidence of inflammation was observed in lungs from uninfected mice. We next wondered whether inhibition of NO by AG treatment would impact levels of inflammatory factors during *N. farcinica* infection. PBS-treated infected mice showed higher levels of TNF-α, IFN-γ, and IL-10 than PBS-treated uninfected counterparts as assessed by ELISA (Figure 3B–D). It is noteworthy that AG treatment significantly reduced the levels of TNF-α, IFN-γ, and IL-10 in infected animals. Overall, AG-treated infected mice exhibited a lower bacterial load and pulmonary inflammatory response.

### 3.2. NO Increases Mortality Rate in N. farcinica-Infected Mice

We then investigated whether AG treatment impacts on long-term consequences of *N. farcinica* infection (14 dpi). The intraperitoneal infection model used in the lethality assay simulates a systemic infection, and there is no doubt that dissemination of the infection occurs, as evidenced by our previous study [19]. This route of infection readily leads to mortality in mice due to multiple organ involvement (e.g., heart, liver, spleen, lung, eye and rectum). Our data showed that PBS-treated infected mice died between 2 and 4 dpi, with a final survival rate of 50% (Figure 4B). By contrast, all AG-treated infected mice survived infection. Our data demonstrated that AG treatment may improve survival rate in infected mice.

### 3.3. NO Contributed to N. farcinica-Induced Inflammation and Bacterial Replication In Vitro

To investigate whether expression of NO might be regulated by *N. farcinica* infection in vitro, we assessed the level of NO by alveolar MH-S macrophages (widely used model cells) using Greiss reaction. The expression of NO has been shown to be higher in *N. farcinica* (multiplicity of infection (MOI) = 50) infected alveolar MH-S macrophages compared with control counterparts at 1, 6, 12 hpi (Figure 5A–C). Upon *N. farcinica* infection at MOI = 50, the level of NO peaked at 12 hpi.

To evaluate the potential role of NO on *N. farcinica* infection, we examined the effect of NO on inflammatory cytokine expression and bacterial replication in alveolar MH-S macrophages. We generated useful experimental models to underexpress NO in target cells (Figure 6A). Inhibition of NO by AG treatment impaired *N. farcinica*-induced expression of TNF-α and IFN-γ as assessed by ELISA (Figure 6B,C). In addition, limitation of bacterial replication was observed in AG-treated, infected alveolar MH-S macrophages than in the control group (Figure 6D). Therefore, it is likely that NO may aggravate *N. farcinica* infection in alveolar MH-S macrophages.

### 3.4. NO Is Induced by N. farcinica Through MAPK JNK and NF-κB Signaling

We investigated whether MAPK signaling regulates the expression of NO. The inhibition of JNK contributed to lower NO expression in infected alveolar MH-S macrophages (Figure 7), suggesting that the expression of NO was dependent on MAPK JNK signaling. The activation of MAPK signaling can contribute to the activation of the downstream signaling NF-κB, inducing inflammatory cytokine production. To explore whether the expression of NO may be regulated by NF-κB signaling, we treated alveolar MH-S macrophages with PDTC for 1 h, respectively. After treatment, alveolar MH-S macrophages were infected with *N. farcinica*, and the expression of NO was detected by Greiss reaction. As shown in Figure 6, we found that the expression of NO was significantly impaired by the inhibition of NF-κB in infected alveolar MH-S macrophages, suggesting that the expression of NO was dependent on NF-κB signaling. Collectively, *N. farcinica* may induce NO expression via MAPK JNK and NF-κB signaling.

## 4. Discussion

*Nocardia* infection poses a challenging threat to human health, especially immunocompromised individuals. The pathogenic mechanism of *Nocardia* remains unclear. In this study, we aimed to explore the impact of NO on *N. farcinica*-infected BALB/c mice and alveolar MH-S macrophages. Our previous research suggested that the mRNA level of iNOS was upregulated in *N. farcinica*-infected BV-2 cells [4]. The expression of NO enhanced after *N. farcinica*-infection, in line with other reports (in mice) [16,18], and remain lower in AG-treated infected mice and alveolar MH-S macrophages compared with AG-untreated, infected counterparts.

Neuronal NOS (nNOS), and endothelial NOS (eNOS) are constitutively expressed cells and tissues, producing small amounts of NO, which participate in important physiological events such as blood flow and blood pressure regulation. AG is a widely used and potent selective inhibitor of iNOS, having a certain inhibitory effect on nNOS and eNOS, but this inhibition is slight, as evidenced by the uninfected group in Figure 2B. Thus, AG is suitable for investigating the effects of inducible NO on *N. farcinica*-infected mice and alveolar macrophages.

After confirming the efficacy of AG, we assessed the effects of NO on *N. farcinica* acute infection. Unlike the findings of and Tarantino et al. (in mice) [20] but in line with the report by Ferreira et al. (in mice) [21], the lower bacterial load was observed in AG-treated infected mice and alveolar MH-S macrophages than AG-untreated, infected counterparts. NO is an unstable free radical with a short lifetime of only seconds, finally oxidized to nitrites and nitrates. *N. farcinica* contains nitrate reductase genes (*narGHIJ* and *nirBD*) [22]. It is plausible to hypothesize that NO could be used by *N. farcinica* for anaerobic respiration, promoting its growth. Jung et al. demonstrated that NO produced by human macrophages is not sufficient to limit intracellular mycobacterial growth, and mycobacteria may use NO to enhance their survival in macrophages [23], powerfully supporting our hypothesis.

Compared to AG-untreated, infected animals, our data suggest reduced inflammation in the lungs of AG-treated, infected animals as assessed by cytokine expression and histopathology. Lungs from AG-treated infected mice showed a significant downregulation of cytokines (TNF-α, IFN-γ, and IL-10) than AG-untreated, infected counterparts, in accordance with other reports [24,25,26]. Similar effects (TNF-α, and IFN-γ) were observed in in vitro experiments. Notably, Okamoto et al. found that *N^G^*-monomethyl-L-arginine (L-NMMA), an iNOS inhibitor, decreased the expression of TNF-α in LPS-stimulated alveolar MH-S macrophages [27]. Additionally, infected tissue from AG-treated animals exhibited a small number of inflammatory cells, in line with the findings of Carmona et al. [16,26].

Then, we explored the effect of NO on *N. farcinica* chronic infection. Our data showed that AG treatment could improve survival rate in infected mice, in line with the findings of Carmona et al. (*N. brasiliensis*) [16]. MacFarlane et al. observed an opposite result in *Salmonella* experimental infection models [18]. We suggest that the differences in AG treatment on survival rate might be due to various strains and immune response.

It was found that *N. farcinica* might upregulate NO expression in a time- and MOI-dependent manner in alveolar MH-S macrophages. Notably, Tsai et al. found that *Propionibacterium acnes* induced NO production in an MOI-dependent manner in RAW264.7 cells [17]. Next, we investigated how *N. farcinica* regulate NO expression in alveolar MH-S macrophages. Our data showed that the inhibition of MAPK JNK and NF-κB P65 might downregulate NO expression. It was suggested that *N. farcinica* may regulate NO expression through MAPK JNK and NF-κB signaling in alveolar MH-S macrophages, in line with the findings of Tsai et al. [17]. The modest effect of JNK inhibition suggests that while the JNK pathway contributes to the inflammatory response, other pathways (such as NF-κB, other MAPKs like p38 or ERK, or possibly the NLRP3 inflammasome) are likely activated simultaneously and can partially compensate when JNK signaling is dampened.

Additionally, there are some limitations in this study. Firstly, we only treated mice and cells with AG before infection, future study is required to assess the impact of the post-treatment of AG on *N. farcinica*-infected mice and cells, aiding the treatment of nocardiosis. Secondly, we primarily focused on the effect of NO on *N. farcinica*-infected mice and cells, ignoring that the cyclooxygenase-2 (COX2) pathway is a crucial parallel inflammatory pathway that often interacts with iNOS. In future research, we will investigate the potential crosstalk between the iNOS/NO and COX2/ProstaglandinE2 (PGE2) pathways. Thirdly, our study only investigated the role of NO in *N. farcinica*-infection in immunocompetent mice. Since *Nocardia* infection predominantly occurs in immunocompromised individuals in clinical settings, future research will focus on exploring the effect of NO in *N. farcinica*-infection using immunodeficient mouse models. Fourthly, AG may have some off-target effects, we will repeat some of the experiments using L-Nil, a specific inhibitor of iNOS in future research. Fifthly, we only treated macrophages with AG for a short time, while a longer pre-incubation period might allow for a more complete inhibition of NO, potentially leading to a stronger suppression of downstream cytokine production. The effect of pre-treating macrophages with AG on cytokine production for a longer duration will be addressed in future research.

## 5. Conclusions

This study highlights the significant effect of NO on the pathogenicity of *N. farcinica* in BALB/c mice and alveolar MH-S macrophages. NO resulted in the pathogenesis of *N. farcinica* in hosts, as reflected by inflammation, bacterial replication, and mortality. Our study will provide a theoretical foundation for understanding the pathogenicity of *N. farcinica* and offers novel insight for developing nocardiosis prevention.

## Figures and Tables

**Figure 1 microorganisms-13-02336-f001:**
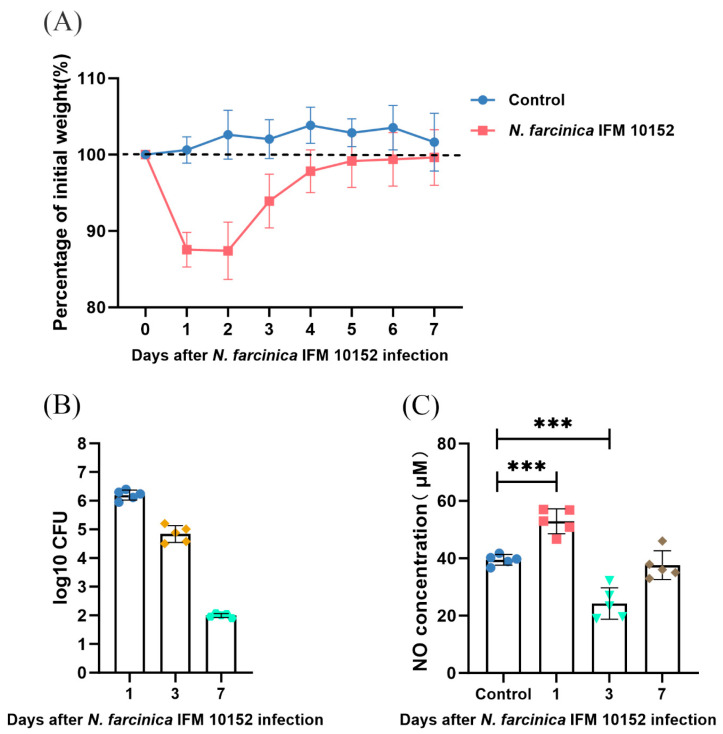
Effects of *N. farcinica* infection on BALB/c mice. (**A**) Change in body weight was serially measured for 7 days (5 mice per group). (**B**) Pulmonary bacterial burden was detected at 1, 3, and 7 dpi by serial dilution (5 mice per time point). (**C**) Expression of NO was assessed at 0, 1, 3, and 7 dpi on basis of Griess reaction (5 mice per time point). The data are presented as means ± SD. *** *p* < 0.001.

**Figure 2 microorganisms-13-02336-f002:**
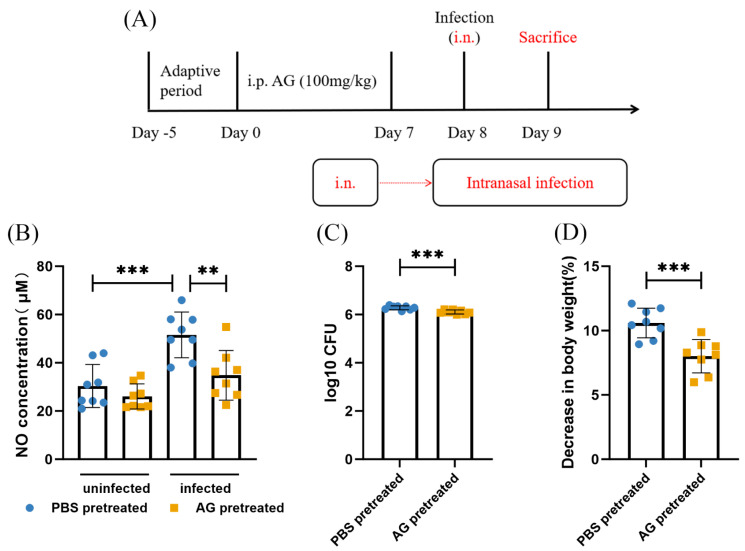
Effects of AG treatment on *N. farcinica*-infected mice at acute phase. (**A**) The experimental flow of inflammatory studies. (**B**) The levels of NO in lung tissues were detected at 24 h post-infection (hpi) by Griess reaction (8 mice per group). (**C**) Bacterial burden in lung tissues was assessed 24 hpi by serial dilution (8 mice per group). (**D**) Change in body weight was recorded before infection and at 24 hpi (8 mice per group). ** *p* < 0.01, *** *p* < 0.001.

**Figure 3 microorganisms-13-02336-f003:**
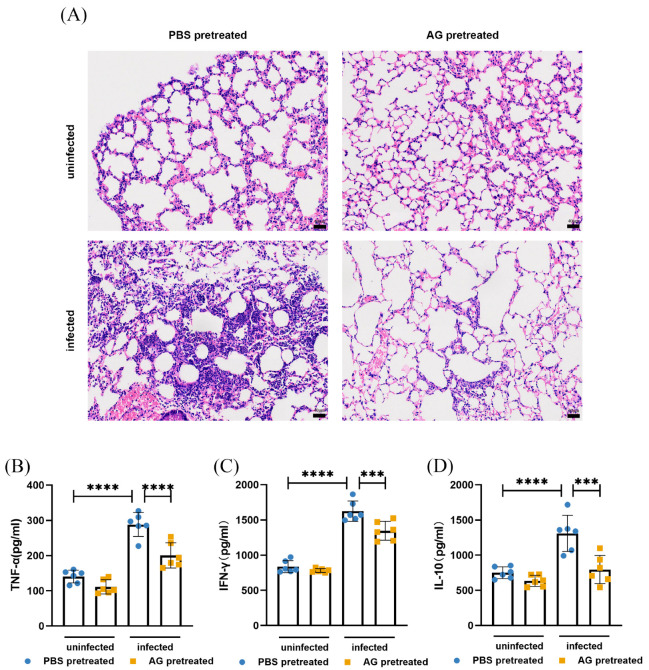
(**A**) Lung inflammation was assessed at 24 hpi by H&E-staining (2 mice per group). Scale bars: 40 µm. (**B**–**D**) Levels of TNF-α, IFN-γ, and IL-10 in lung tissues were measured at 24 hpi by ELISA (6 mice per group). The data are presented as means ± SD. *** *p* < 0.001, **** *p* < 0.0001.

**Figure 4 microorganisms-13-02336-f004:**
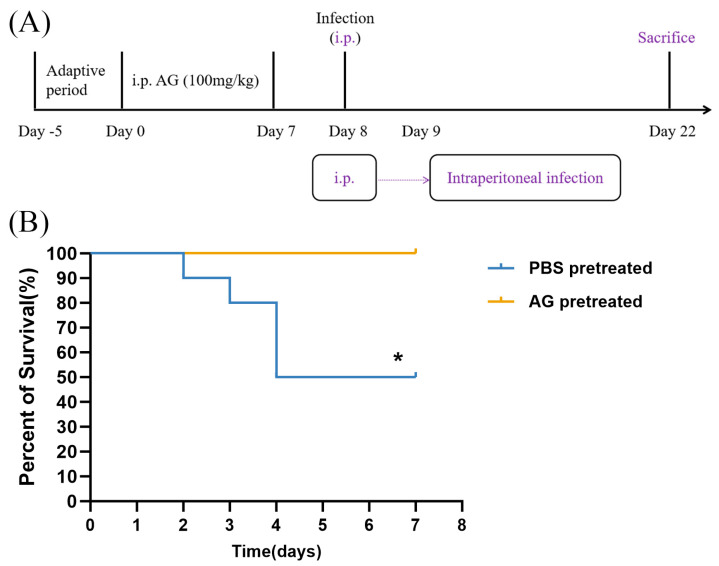
Effects of AG treatment on *N. farcinica*-infected mice at chronic phase. (**A**) The experimental flow of survival study. (**B**) The survival rates of PBS-treated and AG-treated mice with infected *N. farcinica* were serially recorded for 14 days (10 mice per group). * *p* < 0.05.

**Figure 5 microorganisms-13-02336-f005:**
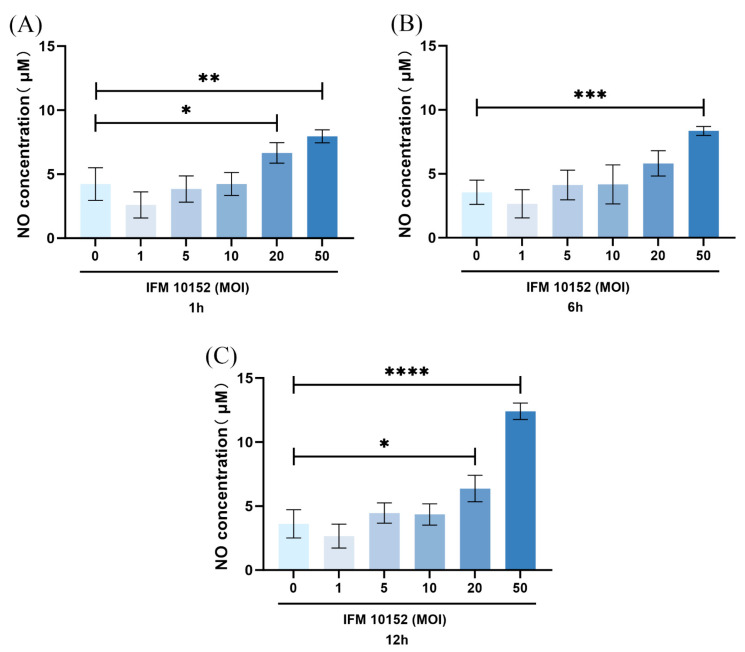
Effects of *N. farcinica* infection on alveolar MH-S macrophages. (**A**–**C**) Alveolar MH-S macrophages were infected with *N. farcinica* at MOI of 0, 1, 5, 10, 20, 50, and the cell samples (3 biological replicates per group per time point) were collected at different time points from 1 to 12 h. The NO expression in alveolar MH-S macrophages was detected on basis of Griess reaction. The data are presented as means ± SD. * *p* < 0.05, ** *p* < 0.01, *** *p* < 0.001, and **** *p* < 0.0001.

**Figure 6 microorganisms-13-02336-f006:**
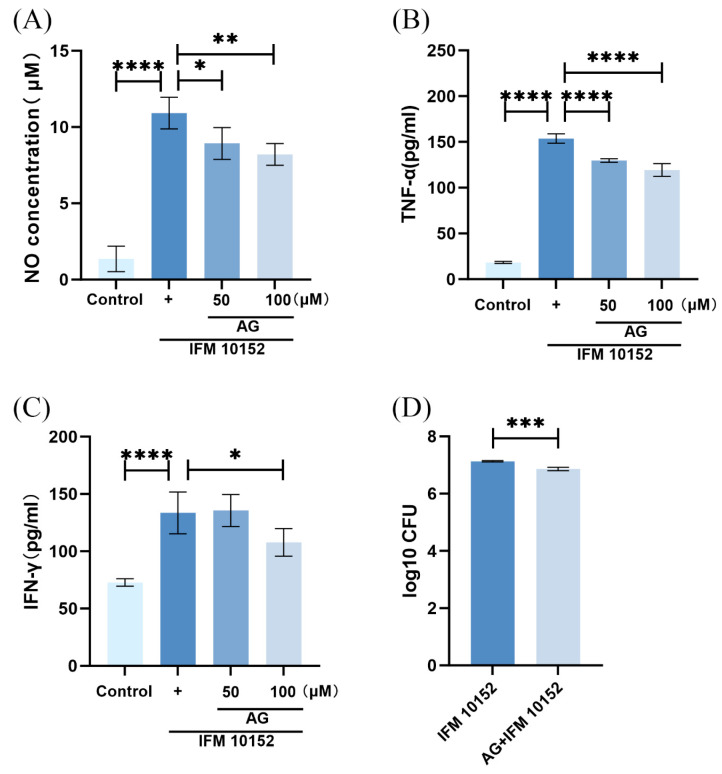
Effects of AG treatment on *N. farcinica*-infected alveolar MH-S macrophages. (**A**–**C**) Alveolar MH-S macrophages were treated for 1 h with or without AG (50 and 100 μM) followed by *N. farcinica* (MOI = 50) stimulation for an additional 12 h (4 biological replicates per group). The levels of NO, TNF-α and IFN-γ were analyzed by Griess reaction and ELISA, respectively. (**D**) Prior to *N. farcinica* (MOI = 50) infection, alveolar MH-S macrophages were treated with or without AG (100 μM) for 1 h. Bacterial replication was detected at 12 hpi by serial dilution (4 biological replicates per group). The data are presented as means ± SD. * *p* < 0.05, ** *p* < 0.01, *** *p* < 0.001, and **** *p* < 0.0001.

**Figure 7 microorganisms-13-02336-f007:**
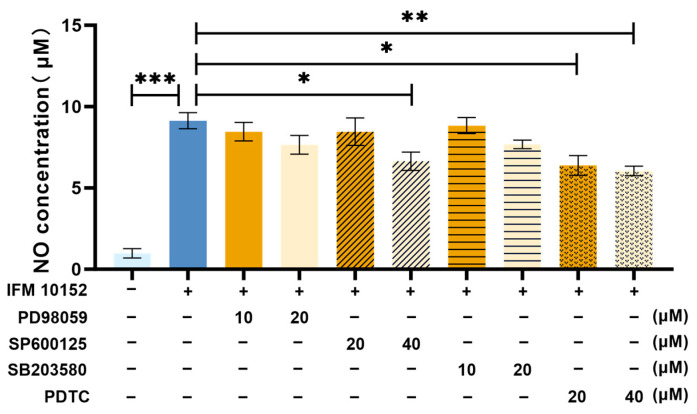
Effects of PD 98059, SP 600125, SB 203580 and PDTC treatment on NO expression in alveolar MH-S macrophages. Alveolar MH-S macrophages were treated for 1 h with or without PD 98059 (10 and 20 μM), SP 600125 (20 and 40 μM), SB 203580 (10 and 20 μM) or PDTC (20 and 40 μM) followed by *N. farcinica* (MOI = 50) stimulation for an additional 12 h. The expression of NO was detected by Griess reaction (4 biological replicates per group). The data are presented as means ± SD. * *p* < 0.05, ** *p* < 0.01, and *** *p* < 0.001.

## Data Availability

The original contributions presented in this study are included in the article. Further inquiries can be directed to the corresponding author, Zhenjun Li, upon reasonable request.

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
