# Peer review of "Nitric Oxide Contributes to the Pathogenesis of Nocardia farcinica Infection in BALB/c Mice and Alveolar MH-S Macrophages"

_microorganisms, 2025, doi:10.3390/microorganisms13102336_

Round 1
Reviewer 1 Report
Comments and Suggestions for Authors
The manuscript is well-written, with only minor English errors. Although novelty is not very high, it provides sufficient new data to warrant its publication.
Minor issues with the manuscript:
- A thorough English read-through
- When harvesting the lungs, it appears that whole lungs were collected and then homogenized in 1 mL PBS. Was homogenization achieved? Other studies have only used either the right or the left components of the lungs.
- Are organisms located only in the lungs or have also disseminated to other organs after 7 days?
- It appears that AG treatment of MH-S macrophages reduces cytokine production after a 1-hour treatment period preceding infection. Could more dramatic effects be obtained with longer incubation times?
- Similarly, inhibitors of the MAPKJNK signaling pathway were effective, but their effect was not very dramatic. What is the author's interpretation of these findings? Other pathways are also involved?
- Why not determine COX2 expression in macrophages?
Reviewer 2 Report
Comments and Suggestions for Authors
They are in the file I uploaded.

Round 2
Reviewer 2 Report
Comments and Suggestions for Authors
The authors have improved the paper, but they misunderstood my objection to using only AMG to inhibit iNOS. AMG works, but it also inhibits NOS1 and NOS3. That is to a problem for the inhibition of cells, but could affect animal survival. That is why I suggested they use L-nil. That should be explained as a limitation since they refuse to do the additional experiment. The papers they site as support for AMG treatment are old and done before the lack of specificity of AMG was known. Those paper also used oral treatment and in my experience mice hate to drink the drug. That is probably why they needed such long pretreatment. This group used i.p injection so they probably did not need a week of pretreatment. I am also not convinced that the small difference in CFU in the lungs explains the mortality. The pathology in the lungs is more clearly shown, but not enough to be lethal. They should have looked for evidence of dissemination.
